# Rheological, Thermal and Mechanical Characterization of Toughened Self-Healing Supramolecular Resins, Based on Hydrogen Bonding

**DOI:** 10.3390/nano12234322

**Published:** 2022-12-05

**Authors:** Liberata Guadagno, Marialuigia Raimondo, Carlo Naddeo, Luigi Vertuccio, Salvatore Russo, Generoso Iannuzzo, Elisa Calabrese

**Affiliations:** 1Department of Industrial Engineering, University of Salerno, Via Giovanni Paolo II, 84084 Fisciano, Italy; 2Department of Engineering, University of Campania “Luigi Vanvitelli”, Via Roma 29, 81031 Aversa, Italy; 3Leonardo Aircraft Division, Viale Dell’Aeronautica, 80038 Pomigliano d’Arco, Italy

**Keywords:** smart materials, mechanical properties, thermosetting materials

## Abstract

This paper proposes the design of toughened self-healing supramolecular resins able to fulfill functional and structural requirements for industrial applications. These new nanocomposites are based on compounds acting as promotors of reversible self-healing interactions. Electrically conductive carbon nanotubes, selected among those allowing to reach the electrical percolation threshold (EPT) with a very low amount of nanofiller, were dispersed in the self-healing polymeric matrix to contrast the electrical insulating properties of epoxy matrices, as required for many applications. The formulated supramolecular systems are thermally stable, up to 360 °C. Depending on the chemical formulation, the self-healing efficiency *η*, assessed by the fracture test, can reach almost the complete self-repairing efficiency (*η* = 99%). Studies on the complex viscosity of smart nanocomposites highlight that the effect of the nanofiller dominates over those due to the healing agents. The presence of healing compounds anchored to the hosting epoxy matrix determines a relevant increase in the glass transition temperature (T_g_), which results in values higher than 200 °C. Compared to the unfilled matrix, a rise from 189 °C to 223 °C is found for two of the proposed formulations.

## 1. Introduction

In the last years, one of the strategic areas of the aeronautics sector regards the demanding challenge to cut fuel consumption and hence CO_2_ emissions in the environment. Since the 1960s, commercial airlines have consistently reduced their CO_2_ emissions by 70%, by controlling fuel employed for aircraft transport. Air traffic is almost as growing as before the COVID-19 pandemic. Therefore, it continues to determine rapidly increasing emissions. The transportation field is responsible for a quarter of CO_2_ emissions caused by humans. Combined strategies to compensate for consumption and environmental pollution can avoid this criticality by becoming even more daunting.

Recently researchers’ attention has been progressively shifting towards new alternative technologies that are less dependent on fossil fuels, bringing electric vehicles to the foreground of the transport sector. Renewable energy source installations may be employed to achieve climate neutrality at reasonable times. For example, wind energy may be relevant in supplying clean energy for recharging board batteries or electrical devices. Innovative cross-cutting solutions are also focused on activities targeted explicitly towards fuel cells and hydrogen, and specific aspects regarding hydrogen storage and its safe use, are under investigation. The selection of technologies should have a holistic approach considering new fuel options and further improvements in the weight reductions of aircraft combined with the increased mechanical performance and durability of the materials and airframe structures.

In this context, the employment of fiber-reinforced composites (FRCs), thanks to their characteristics of lightness and high resistance to fatigue and corrosion, has grown significantly in recent years. New technologies have allowed polymeric composites to compete with metal materials for many applications. In particular, they have been advantageously applied for reducing aircraft weight and, therefore, fuel consumption and emissions of pollutants into the atmosphere. However, a more massive diffusion of these materials requires the resolution of various problems that prevent them from being extended to many parts where their potential could be exploited and maximized. Currently, primary limitations are due to the inability of current composites to fully meet the requirements for aircraft safety certifications in the aviation industry. It is well known that the structural materials of aeronautical aircraft are exposed, in addition to aerodynamic loads, to which they are usually subjected in operating conditions, to adverse atmospheric agents (hail, lightning, storms, strong downbursts, bird impact, etc.). Wind turbines manifest similar vulnerabilities. Severe and prolonged exposures to adverse atmospheric agents, even in recently manufactured structures, can cause substantial damage or trigger micro-cracks, which, spreading, still cause significant damage. The most critical aspect of these phenomena is that the damage generally occurs within the material and is so tiny that it cannot be diagnosed with the naked eye. The currently available detection techniques require periodic monitoring and long inspection times and are not very sensitive. When they detect the damage, it is generally already so severe that it must necessarily intervene with expensive, time-consuming, and complicated repair techniques. The possibility of having self-healing composites can help reduce this criticality. In addition to the auto-repair function, in the field of aeronautical materials, wind turbines, etc., lightweight conductive composites are strongly required to avoid damage from lightning strikes [1,2], to activate anti/deicing and generally for thermal management [3,4,5,6,7] and for being able to confer other functional properties by exploiting conductive nanofillers [8,9,10,11].

The possibility of disposing of self-healing composites can help to reduce this criticality. In addition to this function, in the field of aeronautical materials, electrically conductive lighter composites are strongly required for coffering functional properties to the composites. In the field of self-healing materials, notable progress over the first materials studied has been detected, mostly for rubber, green materials, etc. [12,13,14,15,16,17,18,19,20]. Furthermore, several papers report that nanoparticles can be a valid strategy for creating self-healing composites [21,22,23]. This manuscript deals with the development of multifunctional self-healing supramolecular systems aimed at integrating auto-repair mechanisms in multifunctional structural materials that can contrast the insulating property of epoxy resins. These new nanocomposites are based on molecules acting as self-healing agents and multiwall carbon nanotubes (MWCNTs), which allow for reaching the electrical percolation threshold (EPT) at a lower amount of nanofiller.

In particular, no functionalization of MWCNTs is required (with relevant advantages for possible industrialization processes). Alternative strategies to microencapsulated systems have been recently proposed to auto-repair thermosetting materials. In particular, MWCNTs containing covalently bonded functional groups capable of promoting hydrogen bridging bonds in the hosting polymeric matrix allowed the development of composites with a healing efficiency of around 50–60% [24,25]. It has been demonstrated that the functionalization of CNTs determines an increase in the EPT [25,26] and difficulty in totally inhibiting the CNTs-CNTs assembly, due to the functional groups present on the wall of the CNTs [27]. In this new work, we propose the design of self-healing systems incorporating unfunctionalized MWCNT nanoparticles (able to preserve the electrical properties better) and compounds/molecules having, in their structure, groups able to promote the hydrogen bond formation. These compounds/molecules are compatible with the hosting matrices and can form strong, attractive interactions with the matrix. They are characterized by the intrinsic potentiality to easily create cumulative effects of the reversible interactions, based on hydrogen bonding in the matrix or epoxy network. This solution, combined with unidimensional conductive particles, allows for the manufacturing of multifunctional self-healing epoxy resins. The resulting materials manifest a high healing efficiency, combined with enhanced electrical properties and a good possibility to integrate into the resin self-responsive functions, based on the electrical performance of the material, such as self-sensing, de-icing, and/or anti/icing, based on the Joule effect of the current running through the composite, etc. [7,28,29,30,31,32,33]. In the proposed systems, the incorporation of MWCNTs (or other structured forms of carbon) into the epoxy matrix, is an essential prerequisite to confer the electrical conductivity to the epoxy resin. In fact, the dispersion of CNTs results in a significant increase in the electrical conductivity. The presence of the nanofiller can be beneficial also to increase the amount of interphase (epoxy matrix in contact with the CNT walls) and improve the damping properties, as recently highlighted in the literature [10,34]. For many systems, the presence of this interphase is not only able to mitigate the micro-crack formation, but it is also useful to create a synergic effect with the self-healing fillers.

It is worth noting that the self-healing functionality for the polymeric composites is generally relatively easy to introduce into materials with a high molecular mobility of the chains. Introducing this functionality in the load-bearing materials, without specific external stimuli, is a tough challenge to address, dictated most of all by industrial needs. For “soft” self-healing materials, apart from the high costs (primarily due to the synthesis procedure of non-commercial materials), the problem remains the poor performance in the mechanical properties. The possibility to design structural-functional resins which manifest Tg values around or higher than 190 °C and high healing efficiencies represents a step forward in the industrial applications of these materials, most of all in the aeronautics and aerospace sector, where the extension of the durability in service of this kind of materials allows for saving depletion resources and costs, related to maintenance and standstills of aircrafts (or also of aeolian turbines). Furthermore, the presence of conductive fillers allows for imparting to the resin, other functional properties for damage monitoring, de-icing, etc.

## 2. Materials and Methods

### 2.1. Materials

#### 2.1.1. Choice of Self-Healing Molecules/Compounds

Compounds, characterized by chemical functionalities that establish reversible non-covalent bonds among themselves and with the epoxy matrix, have been selected as healing agents. The structure of these compounds was chosen considering their peculiar property to activate interactions with the epoxy matrix through the formation of reversible hydrogen bonds, generating a supramolecular network with the intrinsic property to repair themselves after damage. It is essential to point out that the structure of the hosting matrix can also contain H-bond “donor” groups, due to the large amount of -OH groups in the cured epoxy matrix. The chemical formulas and the acronyms adopted for the self-healing agents are displayed in Figure 1.

Furthermore, Figure 1 highlights the H-bonding donor and acceptor sites (green and red evidenced, respectively) present in the molecules and which can establish non-covalent interactions with hydroxyl (OH) and carbonyl groups (C=O) of the toughened epoxy matrix. The selected molecules are commercially available powders, and they were purchased from Merck (Merck KGaA, Darmstadt, Germany). They are here named “healing molecules”; they have been dispersed/solubilized into the epoxy matrix before the curing cycle.

#### 2.1.2. Choice and Preparation of the EP Matrix

The epoxy matrix, in which the self-healing molecules were added, is a toughened epoxy resin here named EP and composed of: (a) the terafunctional epoxy precursor tetraglycidylmethylenedianiline; (b) the epoxy reactive diluent 1,4 butanedioldiglycidyl ether; (c) the curing agent 4,4- diaminodiphenyl sulfone; (d) the rubber phase, carboxyl-terminated butadiene acrylonitrile copolymer. The a-c components were provided by Merck (Merck KGaA Darmstadt, Germany), while the d component was purchased from Hycar-Reactive Liquid Polymers. The epoxy matrix, without healing molecules and nanofillers, was prepared as a reference and named sample EP. It was obtained by adding to the epoxy precursor, the rubber phase and the catalyst triphenylphosphine (PPh_3_), in a concentration of 12.5 phr and 10 phr, with respect to the epoxy precursor, respectively. This formulation was treated under mechanical agitation at 170 °C in an oil bath for 24 h. The triphenylphosphine was employed as a catalyst to activate the reaction between the epoxy groups and the carboxylic acid groups of the elastomer. In the first stage, the formation of the phosphonium salt occurs, due to the nucleophilic attack of the triphenylphosphine on the carbon of the epoxy ring. In the second stage, the anion of the carboxylic acid, reacting with the carbon atom of the positive phosphorus, regenerates the catalyst PPh_3,_ simultaneously producing the modified precursor. The reactions allowing this kind of functionalization and hence the epoxy matrix’s toughening, are described in a previous manuscript [10]. The obtained liquid mixture was cooled to the lowest temperature of 120° before adding the reactive epoxy diluent, at a concentration corresponding to a ratio epoxy precursor/reactive diluent of 80/20. Then, the curing agent was incorporated in an amount of 55 phr, with respect to the epoxy precursor, obtaining the liquid epoxy mixture EP, which was treated at 120 °C for 1 h, before degassing in a vacuum at 100 °C for 1 h. The EP formulation was hardened at 125 °C for 1 h, then at 200 °C for 3 h. The same two-step curing cycle was performed for all of the hardened formulations, described hereafter.

#### 2.1.3. Development of the Self-Healing Multifunctional Material

To confer the electrical properties and the auto-repair ability to the toughened epoxy matrix EP, the multiwall carbon nanotubes (indicated by the acronym CNT), and the healing molecules, were added to the epoxy matrix. The CNTs (3100 Grade) were provided by Nanocyl S.A. (Sambreville, Belgium). They were dispersed in the toughened epoxy mixture by ultrasonication for 20 min through a Hielscher model UP200S-24 kHz high-power ultrasonic probe.

The samples with different compositions were prepared. They are listed in Table 1, where the (%) is a weight percentage (*w*/*w*). During the preparation procedure, the healing agent was added to the liquid mixture before incorporating the curing agent, keeping the blend under mixing at 120 °C for 30 min. The MWCNTs were dispersed after the curing agent solubilization. All of the formulated mixtures were degassed under a vacuum and cured by the two-stage curing cycle, as described in the previous section.

### 2.2. Methods

#### 2.2.1. Thermal Investigation

The thermogravimetric analysis (TGA) was performed using a Mettler TGA/SDTA 851 thermal analyzer. A range of temperature from 25 °C to 1000 °C was explored by heating the samples with a speed of 10 °C min^−1^.

The differential scanning calorimetric (DSC) graphs were collected using a Mettler DSC 822/400 (Mettler-Toledo, Columbus, OH, USA). The curing degree (*DC*) of the samples was calculated from the total heat of reaction (Δ*H_T_*) of the cured samples and the residual heat of reaction (Δ*H_Res_*) of the partially cured samples, using Equation (1). A series of isothermal experiments were carried out to obtain the fraction cured at various temperatures. Δ*H_T_* was determined through a dynamic run, resulting in a completely cured resin. The total heat of the reaction was considered, according to Equation (2), where Δ*H_iso_* and Δ*H_Res_* are the areas under the isothermal and dynamic thermograms, respectively.
(1)$$DC=\frac{\Delta H_{T}-\Delta H_{Res}}{\Delta H_{T}}\times 100$$
(2)$$\Delta H_{T}=\Delta H_{iso}+\Delta H_{Res}$$

#### 2.2.2. Dynamic Mechanical Analysis (DMA)

The solid samples with a cuboid geometry (2 × 10 × 35 mm^3^) were solicited with a variable flexural deformation in a three points bending mode (Tritec 2000 DMA-Triton Technology, Mansfield. MA, USA), at the frequency of 1 Hz, a displacement amplitude of 0.03 mm, and between −90 °C to 315 °C at 3 °C min^−1^.

#### 2.2.3. Self-Healing Efficiency Evaluation

The healing efficiency (*η*) of the developed specimens was determined through the fracture tests. In the condition of a quasi-static fracture, the crack healing efficiency *η* is the ability of a healed sample to recover the fracture toughness and is defined by Equation (3):(3)$$\eta =\frac{K_{IChealed}}{K_{ICvirgin}}$$
where *K_ICvirgin_* is the fracture toughness of the initial undamaged sample and *K_IChealed_* is the fracture toughness of the healed sample. Based on the studies performed by Mostoy et al. [35], and following previously published papers [36,37], the healing efficiency is measured by the carefully controlled fracture experiments for both the virgin and the healed sample. These tests, already used successfully in the literature [38,39], utilizes a TDCB geometry, illustrated in Figure 2. The geometry and the adopted procedure guarantee a controlled growth of the crack along the centerline of the sample providing accurate values of the fracture toughness.

The procedure adopted to evaluate *η* involves different steps: (a) a pre-crack is performed on the TDCB undamaged specimen to sharpen the crack-tip; (b) the specimen is tested under displacement control, causing the pre-crack to propagate along the centerline of the sample, until failure. The crack is then closed and allowed to heal at room temperature. Once healed, the sample is tested again until failure. Appendix A shows some pictures regarding the EP-*0.5*CNT-M sample with the TDCB geometry, ready for the self-healing efficiency evaluation test. With this geometry, *η* is obtained through Equation (4), derived from Equation (3), replacing K_IC_ = αP_C_ and considering that for the adopted TDCB geometry α = 11.2 × 10^3^ m^−3/2^.
(4)$$\eta =\frac{P_{Chealed}}{P_{Cvirgin}}$$
where *P_Chealed_* is the critical fracture load (*Pc*) of the healed sample, whereas *P_Cvirgin_* is that of the undamaged (virgin) sample. The fracture tests were performed with an INSTRON mod. 5967 Dynamometer with a load cell of 30 KN and a displacement rate of 0.25 mm/min. The samples were loaded at first failure and then unloaded, allowing the crack face to come back into contact. Following sufficient time (24 h) for the healing efficiency to reach a steady state, the healed samples were tested again. The values of *η* were determined using Equation (4) for the formulations containing the MWCNTs and the self-healing fillers. For all samples, only one cycle was carried out: after the closure of the crack, the samples were kept for 24 h, and the first cycles were performed after the healing of the samples.

Details on the methods and types of equipment employed for the rheological, electrical and self-healing efficiency characterizations, are described in the Appendix A.

## 3. Results and Discussion

### 3.1. Rheological Investigation

One of the engineering aspects associated with self-healing multifunctional formulations loaded with carbon nanotubes is the optimization of the processing steps for their production. In this concern, the study and the control of the rheological properties of the epoxy mixtures are of fundamental importance. The choice of the manufacturing processes and the optimization of the various processing parameters strongly depend on the rheological behaviour of the samples. Generally, for a traditional infusion, very low values of viscosities are required (between 0.3 Pa s and 0.8 Pa s) [40]; for higher values of viscosity, the modified infusion processes can be applied [41]. The approaches, based on the employment of prepregs, are less restrictive, but in any case, when there is the intent to transfer the electrical conductivity property of the CNTs to the resin, a concentration of CNTs beyond the electrical percolation threshold (EPT) is required. For the epoxy systems containing carbon-based nanoparticles, all previous investigations demonstrated that the EPT and the rheological percolation threshold are very similar [7,42]. Hence, the concentrations of CNTs suitable to confer the desired benefits of CNTs to the resin also determine an increase in the viscosity values of the uncured oligomeric epoxy precursors, which may constitute a problem for the manufacturing steps. An appropriate manufacturing process can be chosen if the rheological behaviour is well-known and performed in controlled conditions.

It is worth noting that the curing agent was already included in all of the tested liquid formulations, to analyse the viscosity behaviour in a condition similar to industrial practice. For this reason, temperatures equal to or below 90 °C have been considered to avoid the crosslinking reactions during the rheological tests. The value of 90 °C is well below the temperature value at which the crosslinking reactions start (as shown in Section 3.2).

#### 3.1.1. EP Epoxy Mixture

The rheological behavior of the toughened epoxy matrix EP is shown in Figure 3, where the trend of the complex viscosity (*η**), as a function of the frequency (ω) for the EP blend, obtained through a frequency sweep test in the strain-controlled mode performed at temperatures of 25, 50, 75 and 90 °C, is shown.

In the considered range of the frequency and at each tested temperature, the EP mixture is characterized by a Newtonian behavior with constant values of the complex viscosity, decreasing by about four orders of magnitude, as the temperature increases to 90 °C. The complex viscosity values (*η**) at different temperatures are shown in Appendix A.

#### 3.1.2. EP Epoxy Mixture Loaded with 1% wt. of the Healing Molecule

The trends of the complex viscosity, as a function of the frequency for the EP blends filled with healing molecules, were obtained through the frequency sweep tests in a strain-controlled mode, performed at temperatures of 50, 75, and 90 °C. Figure 4a,b show, respectively, the tests performed at 50 and 90 °C and highlight that the inclusion of the self-healing agents does not influence the Newtonian behavior of the matrix EP. Furthermore, the toughened epoxy matrix EP, with the addition of the self-healing agents DBA, M, and T, is characterized by an increase in viscosity, compared to the EP blend.

The comparison between Figure 4a,b evidences a decrease of the complex viscosity as a function of temperature, as expected. In more detail, the viscosity decreases as the temperature increases by about four orders of magnitude for the EP-DBA blend (at 90 °C), of about five orders of magnitude for EP-T blend (at 90 °C) and of about three orders of magnitude for EP-M blend (at 90 °C). The values of the complex viscosity (*η**) at different temperatures for the EP blend, loaded with healing molecules, are displayed in Appendix A.

#### 3.1.3. EP Epoxy Mixture Loaded with 0.5% wt. of CNT

The inclusion of 0.5% wt. CNT in the toughened epoxy matrix EP, determines a change in the viscosity trend, which begins to show a shear-thinning behavior already at T = 25 °C. This trend is more evident at higher temperatures (see Figure 5a). Therefore, the addition of the nanotubes involves the transformation from a Newtonian behavior to a non-Newtonian behavior. Moreover, the inclusion of the 0.5% wt. CNT, in the EP epoxy matrix, produces a plateau in the storage modulus (G’) at low frequencies, indicative of a clear solid-like behaviour (see Figure 5b), highlighting that the EP-*0.5*CNT liquid dispersion is above the rheological percolation threshold, as also confirmed by the electrical results in Section 3.5 of Results and Discussion.

#### 3.1.4. EP Epoxy Mixture Loaded with 1% wt. of the Healing Molecule and 0.5% wt. of CNT

The inclusion of self-healing agents in the mixture EP-*0.5*CNT does not significantly alter the rheological behavior in terms of viscosity and storage modulus trends (see Figure 6 and Figure 7). In particular, at the temperature of 25 °C and in all of the analysed ranges of frequencies, the complex viscosity shows a shear thinning behaviour. It is characterized by higher values than those corresponding to the nanocomposite, containing only 0.5% by weight of CNT (see Figure 5). At the temperature of 50 °C, the complex viscosity shows a shear-thinning trend, much more marked than that occurring at 25 °C (see Figure 6b). Furthermore, in the range of the high frequencies, the mixture containing the barbituric filler DBA, shows the highest viscosity. At temperatures of 75 °C and 90 °C, in the range of high frequencies, this difference in the viscosity values between the various analysed systems, tends to decrease (see Figure 6c,d).

The analysis of the storage modulus as a function of the oscillation frequency highlights that when the temperature increases, a plateau in the area of the low oscillation frequencies is observed for all of the nanocomposite systems. In particular, at the temperature of 50, 75 and 90 °C, the materials are above the rheological percolation threshold (see Figure 7b–d).

#### 3.1.5. EP Epoxy Mixture Loaded with 1% wt. of the DBA Healing Molecule and 0.1, 0.3, 0.5 and 1.0% wt. of CNT

The rheological tests at different temperatures have been performed on the multifunctional self-healing formulations, to evaluate the rheological percolation threshold. The mixture EP, containing the barbituric filler DBA with varying concentrations by weight of the MWCNTs, was investigated. This compound is completely soluble at 1% by weight within the EP epoxy matrix. Increasing the amount of MWCNTs from 0.1 to 1% wt., within the EP-DBA matrix, the complex viscosity in the low frequencies area gradually increases, highlighting a shear thinning behaviour. The complex viscosity trend modifies from a Newtonian to a shear-thinning behavior when the MWCNT’s content is about 0.3% wt. (see Figure 8a).

Analogously, the analysis of the storage modulus shows that the inclusion of the nanofiller gradually modifies the terminal liquid-like behaviour of the pure matrix.

In particular, at a temperature of 25 °C, the system shows a plateau for the storage modulus, in the area of the low oscillation frequencies when the concentration of carbon nanotubes is 1% by weight (see Figure 9a). At a temperature of 50 °C, in the area of the low oscillation frequencies, it is possible to observe, already for the nanocomposite containing 0.5%, by weight of the carbon nanotubes, a plateau value for the elastic modulus (see Figure 9b). Considering the higher temperatures of 75 and 90 °C, the presence of a plateau for G’ is observed already for the percentage of 0.3% by weight of the carbon nanotubes (see Figure 9c,d), indicating, for this value, the rheological percolation threshold. Therefore, the transition from liquid-like to solid-like behaviour, due to the formation of an interconnected network in the nanocomposite, occurs between 0.1 and 0.3% wt. MWCNT.

In conclusion, the rheological measurements carried out on the epoxy EP system, have shown that the presence of the fillers at 1% wt. does not alter the Newtonian behaviour of the epoxy matrix. The addition of the carbon nanotubes, in the epoxy formulations containing the healing agents, shows that the complex viscosity and the storage modulus are mainly dominated by the presence of the nanofillers. The addition of the different amounts of MWCNTs to the EP-DBA mixture, have highlighted that the rheological percolation threshold is between 0.1 and 0.3%, by weight of the MWCNT, similarly to what has been previously observed for the nanocomposite without the acid-based barbituric filler [7].

### 3.2. Thermal Analyses: Thermogravimetric Analysis (TGA) and Differential Scanning Calorimetry (DSC)

Figure 10 shows the results of the thermal analyses. In particular, Figure 10a,b display TGA graphs of the hardened developed systems: EP sample; EP sample containing CNTs (0.5% wt.), EP sample with both CNTs and the self-healing fillers, in air (see Figure 10a) and nitrogen flow (see Figure 10b). The thermal degradation starts around 360 °C for EP-0.*5*CNT-DBA, both in the air and nitrogen atmosphere. Appendix A. The section shows the main parameters of the TGA investigation: Td_5%_, Td_50%_, and the residue at 900 °C, where Td_5%_ and Td_50%_ are the values of temperature corresponding to the mass loss of 5% wt. and 50% wt., respectively. The temperature corresponding to the initial value of the thermal degradation has been read in the TGA graph in correspondence to the 5% mass loss (Td_5%_), as already adopted in the literature [26]. All collected results highlight that the developed functional samples are thermally stable in air and nitrogen, up to temperatures of 316–360 °C (see Appendix A). In all TGA graphs, including the EP matrix, both in an inert atmosphere and air, the first main step of the thermal degradation occurs in the temperature interval from ~360 °C to ~450°C, evidencing that the self-healing fillers and nanotubes hardly affect this step of degradation.

As already seen for the systems loaded with the functionalized CNTs [24], the samples manifest, in this range of temperature, a better thermal stability in air; the mass loss is 52–57% in air and 69–76% in the inert environment, except for the system containing the murexide, for which the mass loss is about 60% in the nitrogen flow. In the inert atmosphere, the nanotubes determine an enhancement in the thermal stability in the second step of the thermal degradation, between 450 °C and 750 °C. Except for the M filler, the other self-healing fillers do not change this trend; due to the CNTs, the DBA compound tends to decrease this effect minimally, and the filler M highly increases it up to 700 °C.

A DSC investigation was carried out to evaluate the *DC* (Equation (1)) of the hardened formulations. Figure 10c,d show the DSC graphs of the uncured and cured samples, respectively. The DSC curves of Figure 10d evidence the exotherms in the range 175–280 °C, due to a small fraction of uncured resin. The *DC* of sample EP is 97%. The inclusion of CNTs in the samples results in a decrease in the value of *DC* (*DC* = 89%), due to a lower crosslinking density; whereas for the sample loaded with both, the CNTs and the barbiturate fillers DBA and T, only a very slight decrease is detected in the values of *DC*, with respect to the value of 97% related to the matrix (see Table 2). Instead, for the epoxy nanocomposite filled with the filler M, a further decrease in the *DC* value, with respect to the unfilled formulation, is detected (*DC* = 86%).

This decrease is likely due to a chemical interaction between the matrix and the murexide molecule, whose oxide anion can induce the opening of the terminal oxirane ring in the epoxy precursor. This results in the chemical binding of the murexide molecule with the polymer matrix, which can hinder the polymerization reaction, leading to a decrease in the curing degree. It is worth noting that the specific self-healing fillers DBA and T determine a higher curing degree, compared to the sample containing only CNTs. Hence, these fillers also help to meet the industrial requirements related to the minimum value of DC (not less than 90% for the structural materials).

### 3.3. Dynamic Mechanical Analysis (DMA)

The DMA studies were carried out to understand whether the fillers responsible for the self-healing affect T_g_ and the storage modulus of the formulated hardened samples.

Figure 11 shows the results of the DMA analyses carried out on the hardened samples. The comparison among the graphs, displayed in Figure 11a, highlights that, except for the sample containing the DBA filler, the presence of CNTs, embedded in the EP matrix, causes a shift of the glass transition temperature (T_g_) to higher values, from 190 to 216 °C (see Figure 11b). The filler T and M determine the higher increase in the Tg of the samples (216 and 214 °C, respectively). An increase of the storage modulus in the temperature range from −90 °C to 0°C (see the black curve of Figure 11c), is also detected for the sample containing only CNTs. For temperatures higher than 0 °C, the difference between EP and EP-*0.5*CNT becomes negligible (see Figure 11d). The joint presence of CNT and the self-healing fillers causes a decrease in the storage modulus, in the whole range of the temperature, before the drop of the storage modulus, at around 150 °C).

### 3.4. Self-Healing Efficiency Evaluation Test

The healing efficiency values (*η*) for the formulated samples have been calculated, according to Equation (4), as described in Section 2.2.3. Self-healing efficiency evaluation.

Figure 12a–c show the load-displacement curves for the EP epoxy matrix loaded with 0.5% wt. of CNT and 1% wt. of fillers DBA (see Figure 12a), T (see Figure 12b) and M (see Figure 12c), and Figure 12d reports a histogram showing the healing efficiency values for the tested samples. The good values of the healing efficiency allow us to suppose a synergistic action between the various components which, interacting with each other into the mixture, could generate a supramolecular network able to activate the self-healing mechanisms.

To confirm this hypothesis, a complete set of TDCB fracture tests has been performed on the reference samples, loaded only with CNT (EP-0.*5*CNT) or the self-healing molecule (EP-DBA, EP-T and EP-M). The healing efficiency and the Pc values for the virgin and healed reference samples are reported in Table 3.

The EP-*0.5*CNT-M composite completely recovers the damage, and the highest efficiency value obtained for this sample could be due to the chemical nature of the molecule. In fact, as depicted in Figure 1, the M filler is characterized by more hydrogen bonding donor and acceptor sites than the other two compounds. In addition, it presents sites that can establish ionic interactions.

It is worth noting that a minimal and intrinsic ability to the auto-repair is always manifested by the epoxy matrix, due to the high amount of -OH groups formed during the crosslinking reactions responsible for the solidified structure. The -OH groups can also exert reversible hydrogen interactions in the polymerized epoxy resins (cured with primary aromatic amines, such as the DDS). However, the presence of CNTs, in the absence of self-healing fillers, determines an increase in the healing efficiency of up to 15%. The simultaneous incorporation of CNTs and the self-healing fillers in the epoxy matrix determines a synergic effect in imparting a self-healing function that reaches values higher than 65%.

### 3.5. Electrical Results

Electrical conductivity measurements have been performed to evaluate the electrical behaviour of nano-charged materials. The obtained values of the electrical conductivity are shown in Table 4. These results have highlighted that the presence of the self-healing fillers in the nano-filled epoxy matrix does not decrease its electrical conductivity. The conductive paths in the nanocomposite are not affected by the addition of the self-healing fillers and the good electrical conductivity values, together with the promising results of the self-healing efficiency tests, confirm the success of this strategic approach, aimed at obtaining a multifunctional load-bearing material with the auto-repair ability.

The detected values highlighted that the self-healing function might be easily combined with other desired functionalities, such as the easy thermal management, the self-sensing property [43,44], etc. It is worth noting that the multifunctional self-healing systems containing the same amount of CNTs, functionalized with hydrogen bonding moieties (to activate self-healing mechanisms), manifest a higher EPT because the functionalization procedure of the nanofiller reduces its electronic performance.

## 4. Conclusions

In this paper, the approach developed for designing autonomous functional self-healing thermosetting load-bearing resins, involves the incorporation of self-healing molecules directly in the epoxy precursors, modified with rubber domains able to impart a local molecular mobility to the resin without decreasing the main glass transition temperature. The dispersion in the resin of the electrically conductive CNTs allows to prepare the resin to perform the functional properties related to the electrical properties of the formulated nanocomposites. In this context, the aspects related to the processing parameters, such as the control of the rheological properties of the epoxy mixtures, have been analysed.

The developed self-healing epoxy resin-based formulations are characterized by high values of T_g_, ranging from 187 to 214 °C, and storage modulus values between 5000 and 2000 MPa up to 60 °C. The presence of CNTs allows for contrasting the insulating properties of the unfilled matrix, bringing the value of the electrical conductivity from 1.16 × 10^−14^ to 2.56 × 10^−2^ S/m. The self-healing fillers DBA and M, in the nano-filled epoxy matrix do not decrease its electrical conductivity, compared to the matrix containing only unfunctionalized CNTs. The conductive paths in the nanocomposite are not affected by the addition of the self-healing fillers and the good electrical conductivity values, together with the promising results of the self-healing efficiency tests, confirm the strategic approach proposed. Furthermore, the electrical conductivity value detected for the self-healing samples points to new scenarios in the multifunctional structural materials. Studies on the rheological behaviour evidence that the presence of the filler at 1% wt. does not alter the Newtonian behaviour of the epoxy matrix. The addition of CNTs, in the epoxy matrices containing the healing compounds, shows that the presence of the nanofiller dominates the complex viscosity and the storage modulus.

Results from the DMA evidence that the CNTs dispersed in the matrix determine an increase in the storage modulus up to 0 °C, leaving the value unchanged, compared to the matrix up to about 150 °C. The simultaneous presence of CNT and the self-healing fillers T and M determines a shift in the peak of Tan δ, which moves towards the higher temperatures (higher than 200 °C), compared to the peak corresponding to the main transition at 190 °C of the unfilled sample. In particular, an increase from 189 °C to 223 °C is detected for the EP-*0.5*CNT-T sample; whereas the increase from 189 °C to 213 °C is observed for sample EP-*0.5*CNT-M. The barbituric filler DBA shows a different behaviour, but also in this case, it is possible to observe a value of Tan δ, almost equal to the unfilled matrix.

## Figures and Tables

**Figure 1 nanomaterials-12-04322-f001:**
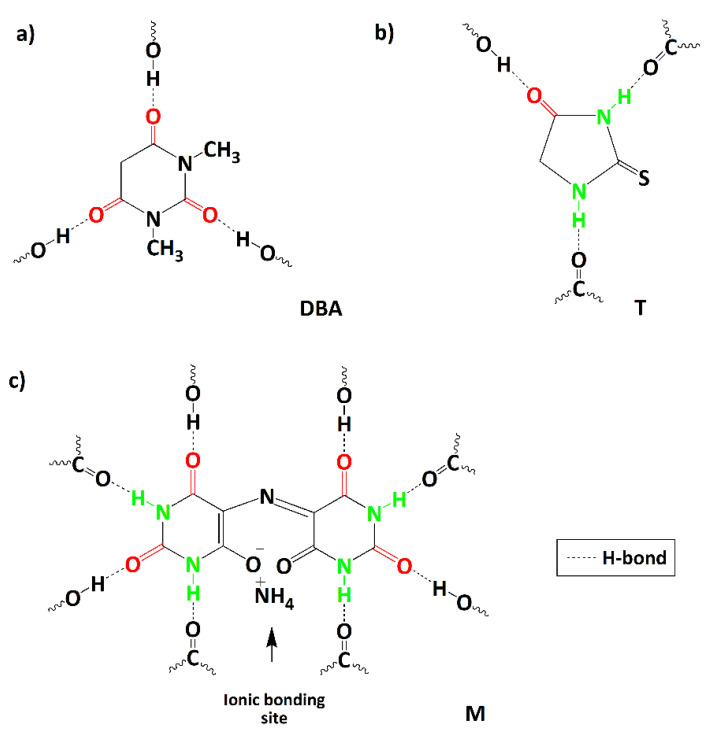
Scheme showing the chemical formulas of the healing molecules and their hydrogen bonding donor and acceptor sites: (**a**) 1.3-Dimethylbarbituric acid (DBA), (**b**) 2-Thiohydantoin (T), and (**c**) Murexide (M).

**Figure 2 nanomaterials-12-04322-f002:**
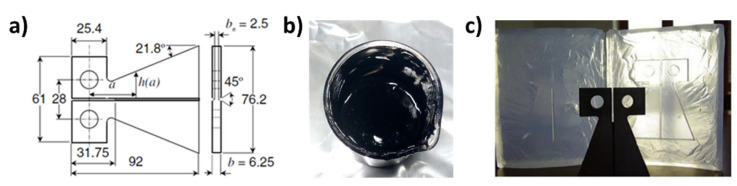
(**a**) TDCB geometry and dimensions (mm); (**b**) prepared mixture; (**c**) molded sample.

**Figure 3 nanomaterials-12-04322-f003:**
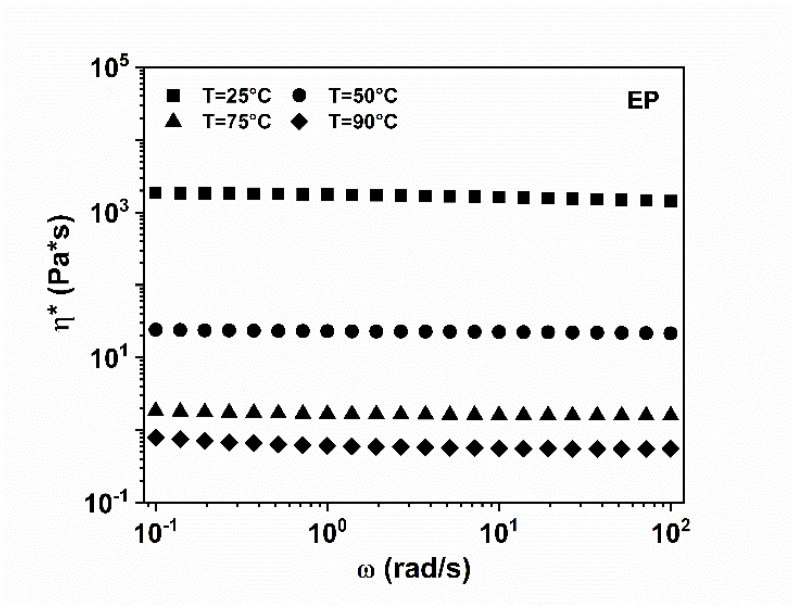
Complex viscosity (*η**) vs. frequency (ω) for the EP blend at different temperatures.

**Figure 4 nanomaterials-12-04322-f004:**
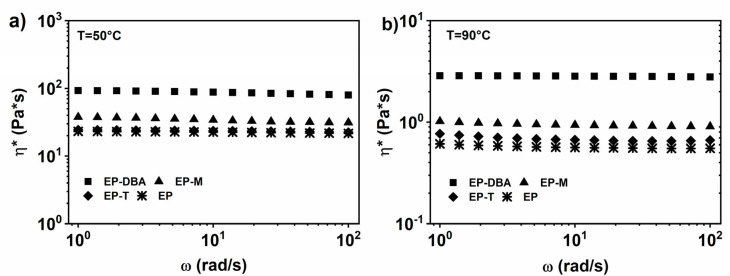
Complex viscosity (*η**) vs. frequency (ω) for the EP, EP-DBA, EP-M and EP-T epoxy mixtures, at: (**a**) T = 50 °C; (**b**) T = 90 °C.

**Figure 5 nanomaterials-12-04322-f005:**
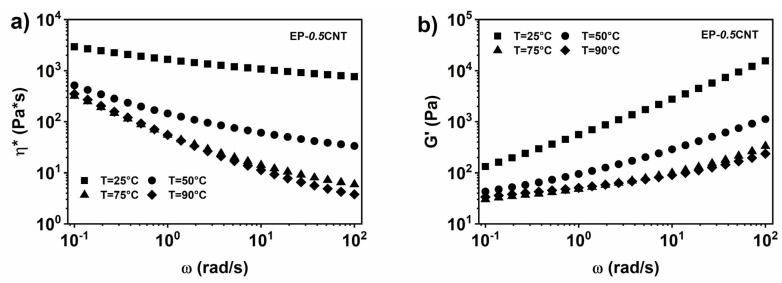
(**a**) Complex viscosity (*η**) vs. frequency (ω) for EP-*0.5*CNT, at different temperatures; (**b**) storage modulus (G’) vs. frequency (ω) for EP-*0.5*CNT, at different temperatures.

**Figure 6 nanomaterials-12-04322-f006:**
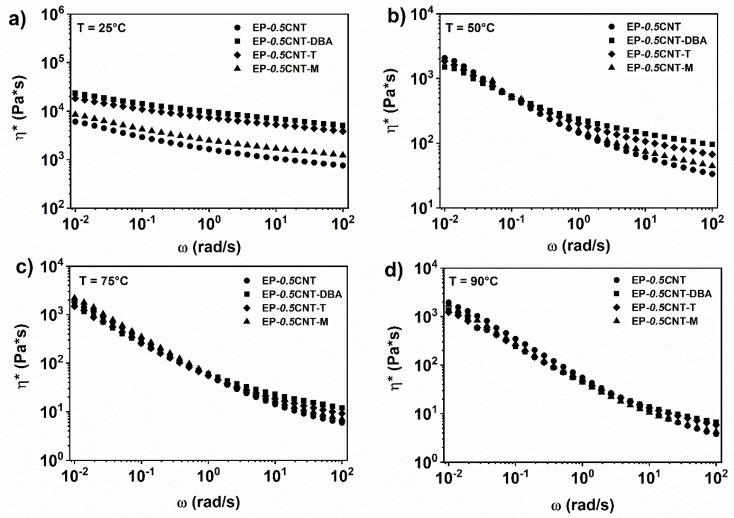
Complex viscosity (*η**) vs. frequency (ω) for EP-*0.5*CNT, EP-*0.5*CNT-DBA, EP-*0.5*CNT-T and EP-*0.5*CNT-M at (**a**) T = 25 °C; (**b**) T = 50 °C; (**c**) T = 75 °C; (**d**) T = 90 °C.

**Figure 7 nanomaterials-12-04322-f007:**
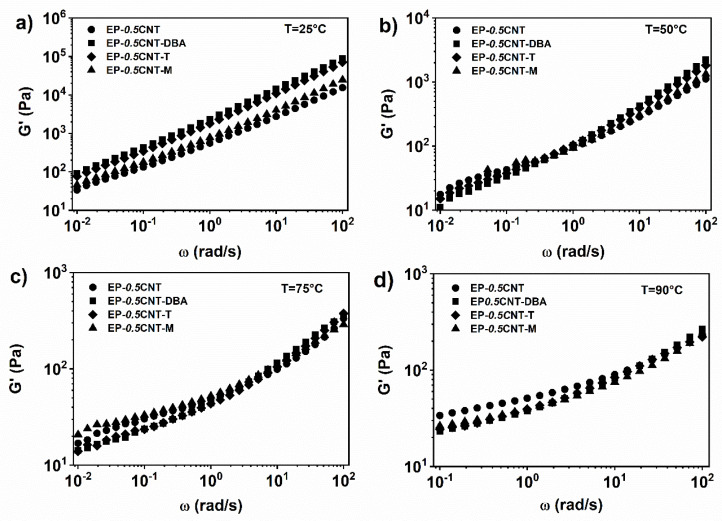
Storage modulus (G’) vs. frequency (ω) for EP-*0.5*CNT, EP-*0.5*CNT-DBA, EP-*0.5*CNT-T and EP-*0.5*CNT-M at (**a**) T = 25 °C; (**b**) T = 50 °C; (**c**) T = 75 °C; (**d**) T = 90 °C.

**Figure 8 nanomaterials-12-04322-f008:**
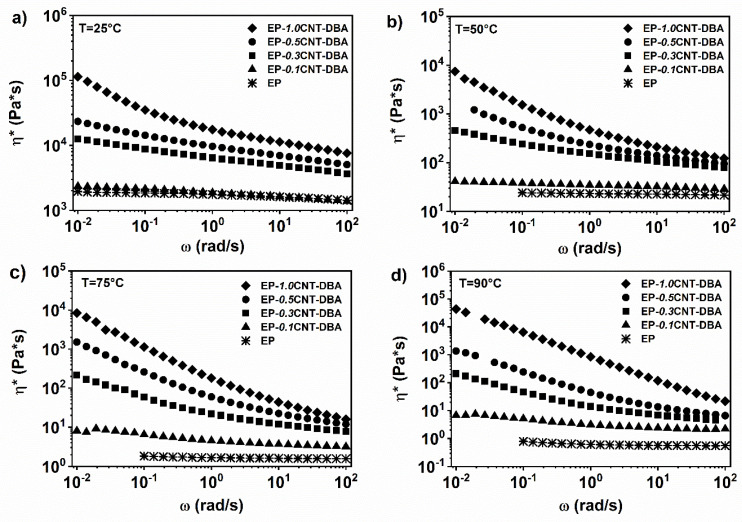
Complex viscosity (*η**) vs. frequency (ω) for EP-DBA with a different concentration of CNT at: (**a**) T = 25 °C; (**b**) T = 50 °C; (**c**) T = 75 °C; (**d**) T = 90 °C.

**Figure 9 nanomaterials-12-04322-f009:**
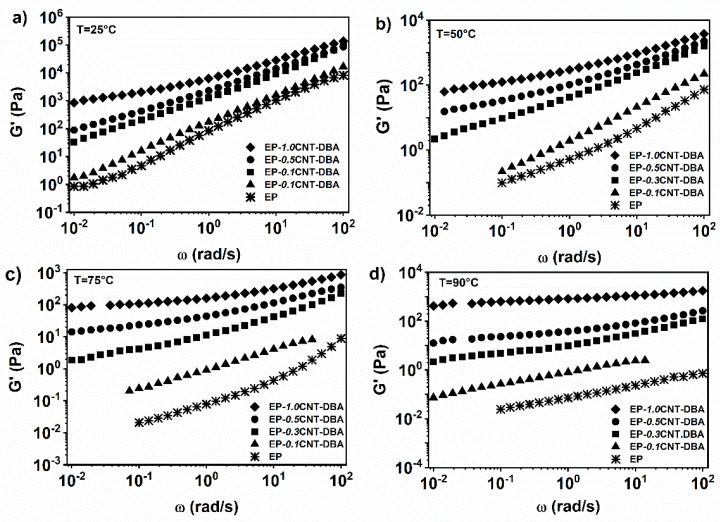
Storage modulus (G’) vs. frequency (ω) for EP-DBA with different concentrations of MWCNTs at: (**a**) T = 25 °C; (**b**) T = 50 °C; (**c**) T = 75 °C; (**d**) T = 90 °C.

**Figure 10 nanomaterials-12-04322-f010:**
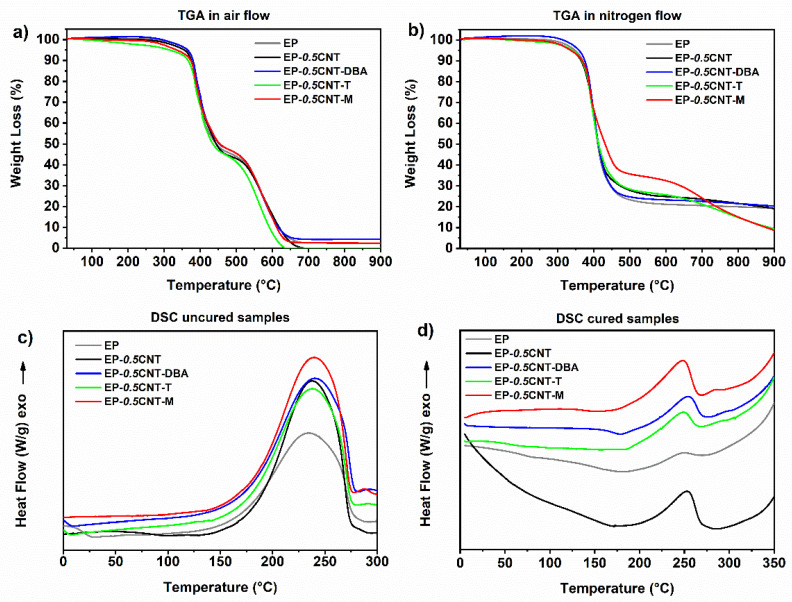
TGA curves of the oven cured epoxy formulations: (**a**) in air and in (**b**) nitrogen flow; (**c**) DSC curves of the raw epoxy formulations; (**d**) DSC curves of the cured epoxy formulations.

**Figure 11 nanomaterials-12-04322-f011:**
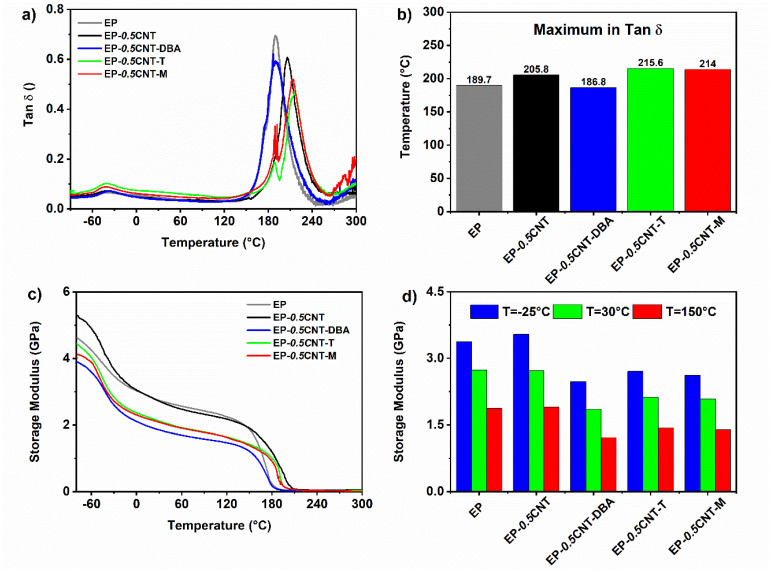
Results from the DMA analyses of the analyzed samples: (**a**) Tan δ curves as a function of temperature; (**b**) Temperatures of the maximum in Tan δ (**c**) storage modulus as a function of temperature; (**d**) storage modulus at different temperatures (−25, 30 and 150 °C).

**Figure 12 nanomaterials-12-04322-f012:**
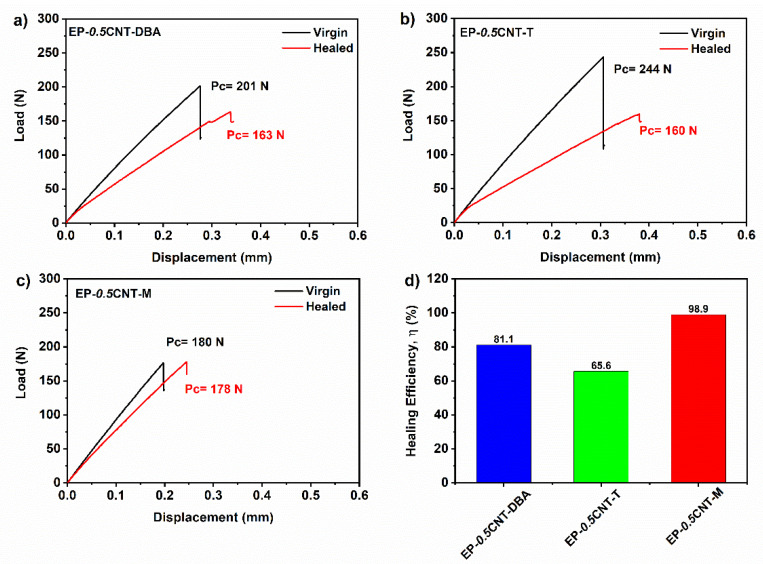
Load displacement curves for the samples: (**a**) EP-*0.5*CNT-DBA, (**b**) EP-*0.5*CNT-T and (**c**) EP-*0.5*CNT-M and (**d**) histogram reporting the healing efficiency values for the tested samples.

**Table 1 nanomaterials-12-04322-t001:** Composition and abbreviations of the prepared samples.

Sample	CNT (%)	Healing Molecule (%)
EP	0	0
EP-DBA	0	1
EP-T	0	1
EP-M	0	1
EP-*0.5*CNT	0.5	0
EP-*0.5*CNT-DBA	0.5	1
EP-*0.5*CNT-T	0.5	1
EP-*0.5*CNT-M	0.5	1
EP-*0.1*CNT-DBA	0.1	1
EP-*0.3*CNT-DBA	0.3	1
EP-*1.0*CNT-DBA	1.0	1

**Table 2 nanomaterials-12-04322-t002:** DSC data of the cured samples.

Sample	Cure Degree *DC* [%]	Δ*H_Res_* [Jg^−1^]	Δ*H_T_* [Jg^−1^]
EP	97	8.16	283.42
EP-*0.5*CNT	89	48.48	429.32
EP-*0.5*CNT-DBA	92	30.10	354.80
EP-*0.5*CNT-T	92	27.22	348.32
EP-*0.5*CNT-M	86	54.27	392.17

**Table 3 nanomaterials-12-04322-t003:** Data of the healing efficiency tests of the tested samples.

Sample	*Pc_virgin_* [N]	*P_Chealed_* [N]	Healing Efficiency (*η*) [%]
EP-*0.5*CNT	525	79.0	15.0
EP-DBA	475	57.0	12.0
EP-T	415	79.0	19.0
EP-M	349	19.6	5.50
EP-*0.5*CNT-DBA	201	163	81.1
EP-*0.5*CNT-2T	244	160	65.6
EP-*0.5*CNT-M	180	178	98.9

**Table 4 nanomaterials-12-04322-t004:** Values of the electrical conductivity for the analyzed samples.

Sample	Electrical Conductivity [S/m]
EP	1.16 × 10^−14^
EP-*0.5*CNT	2.56 × 10^−2^
EP-*0.5*CNT-DBA	1.15 × 10^−2^
EP-*0.5*CNT-T	2.27 × 10^−4^
EP-*0.5*CNT-M	1.29 × 10^−2^

## Data Availability

Not applicable.

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
