# Peer review of "Rheological, Thermal and Mechanical Characterization of Toughened Self-Healing Supramolecular Resins, Based on Hydrogen Bonding"

_nanomaterials, 2022, doi:10.3390/nano12234322_

Round 1

Reviewer 1 Report

The MS is very descriptive and provide very little thorough discussion. It appears more like a technical report.

Also the authors should avoid self-citation as about 50% of the references are from the first author.

Of utmost importance the authors must provide a clear novelty statement with respect to the state of the art and to their previous published papers  

Reviewer 2 Report

In this manuscript, the authors described the synthetic method and related physicochemical properties of hybrid organicinorganic supramolecular resins. In particular, the rheological, thermal, electrical and mechanical properties of such obtained materials have been comprehensively investigated. Overall, the manuscript is well-written and can be accepted in this journal after the following issues are addressed.

1) The title is too general to reflect the specific point(s) in this manuscript, and no research topics is highlighted in such review-like title.

2) As for the self-healing property, to increase the readability, some pictures should be directly taken or even some videos should be added to show such properties in the main text or supporting information.

3) The page number in reference No. 3 was missing.

4) In the abstract, the authors clarified that the propose of this work is to fulfill some requirements for industrial application. However, except the standard characterization, no macroscopic application in the main text or even prospective in conclusion is shown in this work.

5) For real competitive application, the preparation cost/economy of such hybrid composites should be discussed.

Reviewer 3 Report

The authors prepared new conducting epoxy-polymer (EP)/MWCNT (0.1 to 0.5% w/w) composites containing in-matrix-dissolved low-MW additives (1% w/w) capable of forming hydrogen bonds with the aim to provide the composites with self-healing property. Three different commercial low-MW additives were examined and the properties of the composites compared with those of the pure EP, EP/MWCNT and EP/additives reference materials. The composites were characterized by the DSC, TGA and DMA methods, the electric conductivity measurements and their self-healing properties were characterized.

The results obtained are interesting and confirmed the hypothesized self-healing effect of the used low-MW additives. Nevertheless, some correction and improvements should be done.

1.      Hypothesis presented about reaction of murexide with epoxy is not supported by any evidence. Fig. 11 can be therefore omitted as it is enough to write that the oxide anion of a murexide molecule can induce the ring-opening of the oxirane end-group in an epoxy precursor resulting in the chemical binding of the murexide molecule into the polymer matrix.  

2.      Data presented on page 15 (and others) clearly document a strong synergic effect of CNT and low-MW additives on self-healing ability of the resulting polymer. By itself, both types of components provide composites with very poor self-healing capability. However, if both are present in the composite, this capability is quite high. This perhaps the most interesting result is remaining without any attempt to be explained, at least tentatively.

3.      Table 1 should be deleted and abbreviations of additives inserted to Fig. 1. Also Fig. 14 is redundant. Info about H-donor atoms and H-accepting sites can be included in Fig. 1 as H and O.

4.      Symbols of quantities should be italicized.

5.      The term resin is not always used correctly – it should be used for a reactive “oligomeric” precursor but not for the resulting polymer obtained by curing the resin.

6.      rows 437-439 - a little confused sentence.

7.      It would be better to present storage moduli in GPa instead thousands of MPa.

8.      Fig. 13 – the scales of the load vs displacement graphs should have uniform to enable direct comparing.
